# High-Sensitivity Troponin T Testing: Consequences on Daily Clinical Practice and Effects on Diagnosis of Myocardial Infarction

**DOI:** 10.3390/jcm9030775

**Published:** 2020-03-12

**Authors:** Christian Frédéric Zachoval, Ramona Dolscheid-Pommerich, Ingo Graeff, Bernd Goldschmidt, Andreas Grigull, Birgit Stoffel-Wagner, Georg Nickenig, Sebastian Zimmer

**Affiliations:** 1Medizinische Klinik und Poliklinik II, Universitätsklinikum Bonn, 53127 Bonn, Germany; 2Institut für Klinische Chemie und Klinische Pharmakologie, Universitätsklinikum Bonn, 53127 Bonn, Germany; 3Interdisziplinäres Notfallzentrum, Universitätsklinikum Bonn, 53127 Bonn, Germany; 4Klinisches Informationsmanagement, Geschäftsbereich 5, Universitätsklinikum Bonn, 53127 Bonn, Germany

**Keywords:** troponin, high-sensitivity, myocardial infarction, resource utilization

## Abstract

It remains unclear how introduction of high-sensitivity troponin T testing, as opposed to conventional troponin testing, has affected the diagnosis of acute myocardial infarction (AMI) and resource utilization in unselected hospitalized patients. In this retrospective analysis, we include all consecutive cases from our center during two corresponding time frames (10/2016–04/2017 and 10/2017–04/2018) for which different troponin tests were performed: conventional troponin I (cTnI) and high-sensitivity troponin T (hs-TnT) assays. Testing was performed in 18,025 cases. The incidence of troponin levels above the 99th percentile was significantly higher in cases tested using hs-TnT. This was not associated with increased utilization of echocardiography, coronary angiography, or percutaneous coronary intervention. Although there were no changes in local standard operating procedures, study site personnel, or national coding guidelines, the number of coded AMI significantly decreased after introduction of hs-TnT. In this single-center retrospective study comprising 18,025 mixed medical and surgical cases with troponin testing, the introduction of hs-TnT was not associated with changes in resource utilization among the general cohort, but instead, led to a decrease in the international classification of diseases (ICD)-10 coded diagnosis of AMI.

## 1. Introduction

Since their implementation into clinical practice, conventional troponin I (cTnI) assays have enabled physicians to diagnose myocardial infarction more accurately than any other biomarker assay. They have significantly facilitated treatment selection and patient-risk stratification and have become cornerstones of routine clinical practice [1,2,3,4,5]. Because many clinical situations necessitate time-consuming serial sampling to establish the diagnosis of myocardial infarction, the efficiency of cTnI assays has been questioned in the past. The introduction of new, high-sensitivity troponin assays (hs-Tn) aimed to solve the problem by lowering the limit of detection, and thus allowing a more rapid exclusion of myocardial infarction than any conventional or sensitive troponin assay [6,7,8]. However, because troponin is not specific for the etiology of cardiomyocyte death, the assay leaves the clinician with the responsibility to interpret elevated test results in each individual clinical setting, discriminating between different types of myocardial infarction or myocardial injury due to other causes [9,10,11]. Considering the expansion of non-selective troponin testing (e.g., in non-acute coronary syndrome (ACS) situations) and the associated growing number of test results above the 99th percentile in mixed medical and surgical cohorts [12], causal attribution of troponin elevation remains a challenging task, as elevated levels are not uncommon in a variety of different, possibly coexisting, non-cardiac morbidities, where acute coronary artery obstruction is not likely [13,14,15,16,17,18,19,20,21,22,23]. Reliable data and algorithms on high-sensitivity troponin testing and decision making in the emergency care setting, with regard to chest-pain patients or non-ST-elevation ACS, have been published [7,8,24], but no large study has investigated the consequences of using high-sensitivity troponin T (hs-TnT) on the diagnosis of myocardial infarction and resource utilization in an undifferentiated cohort of non-emergency department (ED), non-chest-pain, mixed surgical and medical patients. The present study aimed to test the hypothesis, that introduction of high-sensitivity troponin testing would result in an increase of resource utilization and diagnosis of acute myocardial infarction in an unselected patient population.

## 2. Experimental Section

### 2.1. Study Design

The present work is a retrospective single-center study conducted at a 1300-bed German university hospital. The study was approved by the local ethics committee.

### 2.2. Study Population

We used the hospital’s patient data management system and laboratory database to include in the analysis every case where at least one troponin measurement was obtained during the two corresponding time frames of October to April in an all-comers fashion, regardless of ambulatory or hospitalized status (10/2016–04/2017 for cTnI–LOCI^®^ Cardiac Troponin I Assay, Dimension Vista^®^ 1500, Siemens Healthineers, Eschborn, Germany-and 10/2017–04/2018 for hs-TnT–Elecsys^®^ Troponin T hs, Cobas^®^ e801, Roche Diagnostics, Mannheim, Germany). By definition, two cases could potentially represent the same individual being evaluated in both consecutive time frames, but analysis was restricted to the first individual presentation per assay to avoid double counting. If the patient underwent coronary angiography, the recorded troponin values were restricted to timepoints before the procedure, to exclude elevated values as a result of the intervention. Three hundred sixteen patients with ST-elevation myocardial infarction (STEMI), as well as 595 patients under the age of 18 were excluded. We restricted our analysis to patients of all departments except the cardiac surgery department (906 patients excluded), due to potential problems in data acquisition of external diagnostics (e.g., externally performed coronary angiography or echocardiography). The data collected consisted of age, gender, in-hospital mortality and procedural data, hs-TnT or cTnI values, as well as international classification of diseases (ICD)-10 codes and codes of the German procedure classification (OPS). The following tokens for Germany (ICD-10-GM and OPS) were used for the analysis: German OPS Code 1-275 for coronary angiography (CAG), 8-837 for percutaneous coronary intervention (PCI), 8-98 for admission to the critical care unit (CCU), 8-77 for cardiopulmonary resuscitation (CPR), I21.4 and I21.9 for acute myocardial infarction (AMI) other than STEMI and lastly I25.1 for coronary heart disease (CHD).

Local standard operating procedures regarding diagnosis and coding of AMI as well as national coding guidelines (ICD10-GM) remained the same throughout the study period.

A 99th percentile value for cTnI was considered 0.05 ng/mL and for hs-TnT 14 ng/L, as recommended by the manufacturers.

Subsequently, we analyzed different subgroups: patients hospitalized in the cardiology department (Subgroup 1), patients hospitalized in other departments (Subgroup 2), and lastly, to investigate the findings in a cohort of patients with high a priori probability for the diagnosis of acute myocardial infarction (Subgroup 3), we applied clinically relevant laboratory criteria to all inpatients of the cardiology department in a separate analysis: with high-sensitivity troponin testing based on the ESC hs-TnT 0/1h algorithm [7,8,24,25] and regarding patients with conventional troponin testing with markedly elevated cTnI values (i.e., 5 times the normal upper limit) or a 50% dynamic change between two serial measurements.

### 2.3. Outcome Measures

The primary endpoint was defined as diagnosis of myocardial infarction based on ICD-10 coding. Secondary outcome measures were: total number of troponin values above the 99th percentile, total number of performed echocardiograms, coronary angiographies, percutaneous coronary interventions, admissions to the critical care unit, cardiopulmonary resuscitation, and in-hospital mortality.

### 2.4. Statistical Analysis

Baseline characteristics were described as frequencies and percentages for categorical variables, mean and minimum/maximum, and standard deviation for continuous variables. Categorical variables were analyzed using Pearson’s chi-square or Fisher’s exact tests, and continuous variables with the T-test. A value of *p* < 0.05 was considered significant; *p*-values are two-sided where appropriate. Statistical calculations were performed using IBM SPSS Statistics 25 (IBM Corp. Released 2017. IBM SPSS Statistics for Mac, Version 25.0. Armonk, IBM Corp, New York, NY, USA).

## 3. Results

In total we included 18,025 cases, representing 12,863 individuals with at least one troponin measurement. Of these cases, 9065 (50%) had been tested using the cTnI assay (10/2016–04/2017) and 8960 (50%) cases with the hs-TnT (10/2017–4/2018). Subsequently, we analyzed different subgroups: patients hospitalized in the cardiology department (Subgroup 1), patients hospitalized in other departments (Subgroup 2) and lastly, patients with high a priori probability for the diagnosis of acute myocardial infarction (Subgroup 3).

### 3.1. Overall Cohort

The baseline characteristics of both cohorts were clinically comparable, with the mean age being 61 years in both groups and a slight predominance of male gender (55% for the cTnI as well as for the hs-TnT group; see Table 1). Overall, the analyzed groups differed significantly in the number of troponin results above the 99th percentile (20.1% vs. 46.8%; *p* < 0.001) and below the limit of detection (LoD) (76.9% vs. 15.6%, *p* < 0.001) as well as the number of coded acute myocardial infarction (AMI) (10.1% vs. 6.2%, *p* < 0.001). The rate of outpatient treatment increased significantly from 28.2% to 30.1% (*p* < 0.01). The extent of consecutive testing increased after introduction of hs-TnT from 34.7% to 39.1% (*p* < 0.001).

### 3.2. Inpatients of the Cardiology Department

The mean age of hospitalized cardiology patients was 70 (cTnI) and 71 years (hs-TnT), with a significant increase in male patients since the introduction of hs-TnT (55.9% vs. 59.7%, respectively; *p* < 0.05; Table 2). The quantity of cases with troponin results above the 99th percentile increased significantly (43.9% vs. 75.6%; *p* < 0.001) between the two time frames. There was no significant change in the number of CCU admissions, CPR-rate, or in-hospital mortality. The percentage of patients examined with echocardiography (81.6% vs. 83.2%) or CAG (38.3% vs. 37.1%) as well as the rate of PCI (44.7% vs. 46.6% of patients undergoing CAG) remained stable. The number of coded AMI decreased upon the introduction of hs-TnT (24.9% vs. 19.2%, *p* < 0.001).

### 3.3. Inpatients of Other Departments

Similar to the previously detailed groups, we noticed a significant increase in troponin results above the 99th percentile (18.7% vs. 51.5%, *p* < 0.001) as well as a decrease in diagnosed AMI (4.8% vs. 3%, *p* < 0.001) after introduction of hs-TnT (Table 3). The number of encoded CHD cases decreased slightly with the new assay (11.7% vs. 10.3%, *p* < 0.05).

### 3.4. In-Hospital Patients with High Probability of Myocardial Infarction

Hospitalized patients in the cardiology department with elevated troponin represent the collective with highest probability for the diagnosis of myocardial infarction. In consideration of existing guidelines on non-ST elevation myocardial infarction and to represent daily clinical decision-making, we selected patients with markedly elevated levels of cTnI (i.e., 5 times upper limit of normal, ULN) or a 50% change between two measurements of cTnI or when testing hs-TnT a single value above 52 ng/L or a Delta 0/1 h > 5 ng/L for analysis in this high probability group. In our collective, in both groups 39% of hospitalized patients met these criteria. The frequency of CAG decreased significantly after introduction of high-sensitivity testing (53.4% vs. 46.4%, *p* < 0.001, Table 4) while rates of PCI remained comparable among the two timeframes (55.6% vs. 54.2% of patients undergoing CAG, *p* = 0.027). The significant drop in diagnosed AMI was even more pronounced in comparison to the beforementioned groups (52.3% vs. 30.8%, *p* < 0.001). No significant changes were noticed for echocardiograms, coded CHD or in-hospital mortality.

## 4. Discussion

### 4.1. Troponin above the 99th Percentile and Hospital Admission

In this retrospective analysis of patient data, troponin testing was performed in over 18,000 cases during two corresponding seven-month periods in order to compare a 5th generation sensitive troponin assay (cTnI) with a high-sensitivity troponin assay (hs-TnT; see Table 1). The widespread use of troponin testing in mixed medical and surgical cohorts, even in the absence of symptoms suggestive of ACS, is doubtless the consequence of troponin being a well-documented and effective biomarker for the identification of patients at risk for major adverse cardiac events (MACE) and mortality, even under non-cardiac circumstances [26,27,28,29,30], which can be confirmed with the present study. Regardless of the troponin assay used, the majority of testing was ordered from outside the cardiology department (cTnI 77%; hs-TnT 77%; data not shown) and in a non-ACS setting (ED presentation cTnI 28%; hs-TnT 27%; see Table 1). As expected from previous publications [31,32], the quantity of test results above the 99th percentile increased significantly after the introduction of high-sensitivity troponin assays. With regard to the present overall cohort, we noticed a 2.3-fold increase: 20.1% to 46.8%. As expected, the highest rate of abnormal results for hs-TnT was observed for inpatients of the cardiology department: 75.6%-in comparison to 43.9% with the cTnI assay for the previous time frame. The rate of admission to the hospital decreased significantly from 71.8% to 69.9%. Existing data on chest-pain patients show a 36% decrease in admissions throughout the first 4 years of implementing hs-TnT [33], and to a limited extent, also explain the findings in this study.

### 4.2. Resource Utilization

Despite the previous hypothesis that a rise in abnormal troponin results would trigger further diagnostic or therapeutic modalities, the utilization of different imaging techniques and revascularization procedures did not increase significantly in the overall cohort between the two time frames that were compared: the rate of echocardiography as well as the use of coronary angiography remained stable, independent of the troponin assay applied. The proportion of patients receiving revascularization during coronary angiography slightly increased from 43% to 45%, but this was deemed not significant.

In contrast, among patients with a high probability of being diagnosed with myocardial infarction, we noticed a significant decrease in invasive imaging, showing a reduction in CAG from 53% to 46% while the rate of PCI only slightly decreased (56% vs. 54%). From our data, we cannot determine if the decrease in CAG had an impact on patient treatment and outcome.

Several studies have investigated resource utilization in different settings, e.g., the ED or chest-pain unit, before and after introduction of high-sensitivity troponin testing, giving rather ambiguous results: some of the studies showed increased use [33,34,35,36,37], while others reported unchanged rates of coronary angiography and revascularization [32,38,39,40]. Our results show no significant change in resource utilization and revascularization in the entirety of patients, but document decreased use of invasive imaging in the subgroup of individuals with highest probability of AMI diagnosis. Importantly, local standard operating procedures for inpatient management as well as national recommendations regarding ICD-coding did not change with introduction of high-sensitivity testing at the study site. Although these retrospective data are not sufficient to draw definite conclusions, we believe there is increasing confusion about the distinction of myocardial injury, type 1 and type 2 myocardial infarction in consideration of the rising amount of troponin values above the 99th percentile. This might lead to withholding invasive evaluation of patients despite formally positive algorithms due to trivialization of pathological test results. This finding is supported by the basically unchanged rate of revascularization when CAG is actually performed.

### 4.3. Acute Myocardial Infarction

In a recent analysis of the SWEDEHEART registry, Odqvist et al. showed that the introduction of hs-TnT resulted in an increased diagnosis of myocardial infarction, which was combined with an increased use of coronary angiography and revascularization [41]. Interestingly, there were different observations of this effect within the SWEDEHEART registry itself. When analyzed from a hospital-specific point-of-view, a number of institutions reported a reduction in the rate of AMI, similar to the results in another study, where large inter-hospital differences in the proportion of AMI compared with unstable angina were published [42]. Our present work questions the hypothesis that the quantity of AMI diagnoses increases following the introduction of hs-TnT. Our data show a significant decrease in the diagnosis of AMI in all of the different subgroups. Interestingly, the most pronounced effects were noticed in the high-risk collective (52% vs. 31%), where we also observed a decrease in invasive imaging. Besides the aforementioned trivialization of pathological test results and fewer invasive verifications, we believe that a disregard of type 2 myocardial infarction has increased after introducing high-sensitivity testing. This problem has already been addressed with other assays [43], and may have relevant impacts on ICD-coding, which in its applied version of ICD10-GM, currently provides no possibility to code type 2 myocardial infarction. Furthermore, a meta-analysis of current literature showed distinct variations in classification between type 1 and type 2 myocardial infarction amongst various studies [44], supporting our hypothesis.

The decreased rate of diagnosis of AMI documented in the present study is concerning. Even if the analysis is restricted to cases in the cardiology department with troponin values above the 99th percentile receiving PCI, ICD-coded diagnosis of AMI is significantly lower after introduction of high-sensitivity testing. This observation could have profound impact on in-hospital treatment and future care: misidentification of AMI may lead to detention of oral antiplatelet therapy and other measures of secondary prevention, reduced awareness with ambulatory follow-up visits and ultimately a higher rate of potentially preventable MACE. These findings highlight the urgent need for future prospective analyses on this topic.

### 4.4. CCU Admission, CPR, and In-Hospital Mortality

We noticed no significant change in the rate of admission to the critical care unit or cardiopulmonary resuscitation in all the different subgroups. Receiver operating characteristic curve analysis demonstrated a correlation with hs-TnT and in-hospital mortality with an area under the curve (AUC) of 0.81 (Appendix A). The in-hospital mortality rate did not differ before and after introduction of hs-TnT, which is in line with most of the previously published, observational data for ED or chest-pain / ACS patients [32,34,35]. Only one group of investigators reported a small increase in mortality [33,36], attributed to all-cause mortality, not MACE.

### 4.5. Strengths and Limitations

The present investigation contains data of every inpatient case receiving troponin testing at a large tertiary care hospital. The unselected mixed cohort of patients from different surgical and medical departments reduces the risk of selection bias. To our knowledge, this is the first study analyzing the introduction of hs-TnT in a non-ED, non-chest-pain unit setting with both surgical and medical patients included.

Due to the retrospective approach of the presented study, we used the existing ICD-coding of patient data without further verification or secondary adjudication of diagnoses. The indication for troponin testing was not fully traceable and, to some extent, may be attributable to preoperative risk-stratification directives in the absence of symptoms. The use of coronary angiography and revascularization was not standardized and left to the treating physician, therefore, the true prevalence of significant coronary artery disease in both cohorts still requires further clarification.

## 5. Conclusions

In the present single-center retrospective study comprising 18,025 mixed medical and surgical cases with troponin testing not restricted to the emergency department or ACS setting, the introduction of hs-TnT led to a decrease in the diagnosis of acute myocardial infarction. We noticed significantly reduced use of invasive imaging, restricted to a subgroup of patients with high probability for myocardial infarction, but no change in resource utilization or in-hospital mortality in the overall cohort.

## Figures and Tables

**Table 1 jcm-09-00775-t001:** Demographic and hospital data for all patients in the cohort (cTnI: conventional troponin I; hs-TnT: high-sensitivity troponin T; y: years; SD: standard deviation; Tn: troponin; LoD: limit of detection; ED presentation: first medical contact in the emergency department with cardiovascular complaints; CAG: coronary angiography; PCI: percutaneous coronary intervention; CCU: critical care unit; CPR: cardiopulmonary resuscitation; AMI: acute myocardial infarction; CHD: coronary heart disease; n.s.: not significant; - PCI performed, indicate that PCI performed is a subset of CAG).

	10/2016–04/2017 (cTnI)		10/2017–04/2018 (hs-TnT)		*p*-Value
No. of cases	9065		8960		
Age (y ± SD), (Min-Max)	61 ± 19	(18–102)	61 ± 20	(18–102)	n.s.
Male gender, *n* (%)	4999	(55.1)	4927	(55.0)	n.s.
Tn < LoD (%)	6975	(76.9)	1401	(15.6)	<0.001
Tn > 99th percentile (%)	1818	(20.1)	4197	(46.8)	<0.001
Consecutive Tn testing	3150	(34.7)	3501	(39.1)	<0.001
Outpatients (%)	2555	(28.2)	2695	(30.1)	<0.01
ED presentation	2540	(28.0)	2392	(26.7)	n.s.
Echocardiogram (%)	2745	(30.3)	2631	(29.4)	n.s.
CAG (%)	946	(10.4)	873	(9.7)	n.s.
- PCI performed (% of CAG)	404	(42.7)	395	(45.2)	n.s.
CCU admission (%)	1447	(16.0)	1464	(16.3)	n.s.
CPR (%)	172	(1.9)	172	(1.9)	n.s.
Mortality, *n* (%)	496	(5.5)	494	(5.5)	n.s.
AMI (%)	920	(10.1)	554	(6.2)	<0.001
CHD (%)	1545	(17.0)	1469	(16.4)	n.s.

**Table 2 jcm-09-00775-t002:** Demographic and hospital data for all inpatients of the cardiology department (Subgroup 1). (cTnI: conventional troponin I; hs-TnT: high-sensitivity troponin T; y: years; SD: standard deviation; Tn: troponin; CAG: coronary angiography; PCI: percutaneous coronary intervention; CCU: critical care unit; CPR: cardiopulmonary resuscitation; AMI: acute myocardial infarction, CHD: coronary heart disease; n.s.: not significant).

	10/2016–04/2017 (cTnI)		10/2017–04/2018 (hs-TnT)		*p*-Value
No. of cases	2115		2052		
Age (y ± SD), (Min-Max)	70 ± 15	(18–102)	71 ± 15	(18–102)	n.s.
Male gender, *n* (%)	1182	(55.9)	1226	(59.7)	<0.05
Tn > 99th percentile (%)	929	(43.9)	1552	(75.6)	<0.001
Echocardiogram (%)	1726	(81.6)	1708	(83.2)	n.s.
CAG (%)	810	(38.3)	761	(37.1)	n.s.
- PCI performed (% of CAG)	362	(44.7)	355	(46.6)	n.s.
CCU admission (%)	448	(21.2)	483	(23.5)	n.s.
CPR (%)	91	(4.3)	91	(4.4)	n.s.
Mortality, *n* (%)	115	(5.4)	140	(6.8)	n.s.
AMI (%)	527	(24.9)	393	(19.2)	<0.001
CHD (%)	1012	(47.8)	1019	(49.7)	n.s.

**Table 3 jcm-09-00775-t003:** Demographic and hospital data for all inpatients besides those of the cardiology department (Subgroup 2). (cTnI: conventional troponin I; hs-TnT: high-sensitivity troponin T; y: years; SD: standard deviation; Tn: troponin; CAG: coronary angiography PCI: percutaneous coronary intervention; CCU: critical care unit; CPR: cardiopulmonary resuscitation; AMI: acute myocardial infarction; CHD: coronary heart disease; n.s.: not significant).

	10/2016–04/2017 (cTnI)		10/2017–04/2018 (hs-TnT)		*p*-Value
No. of cases	4395		4213		
Age (y ± SD), (Min-Max)	62 ± 19	(18–98)	63 ± 18	(18–98)	0.049
Male gender, *n* (%)	2491	(56.7)	2333	(55.4)	n.s.
Tn > 99th percentile (%)	824	(18.7)	2170	(51.5)	<0.001
Echocardiogram (%)	825	(18.8)	810	(19.2)	n.s.
CAG (%)	132	(3.0)	109	(2.6)	n.s.
- PCI performed (% of CAG)	41	(31.1)	39	(35.8)	n.s.
CCU admission (%)	999	(22.7)	981	(23.3)	n.s.
CPR (%)	80	(1.8)	81	(1.9)	n.s.
Mortality, *n* (%)	381	(8.7)	354	(8.4)	n.s.
AMI (%)	212	(4.8)	127	(3.0)	<0.001
CHD (%)	514	(11.7)	432	(10.3)	<0.05

**Table 4 jcm-09-00775-t004:** Demographic and hospital data for inpatients of the cardiology department with single cTnI > 0.25 ng/mL (= 5 × upper limit of normal) or 50% dynamic change between two measurements in the timeframe from 10/2016 to 04/2017 (cTnI 5 × ULN/50% delta) and hs-TnT ≥ 52 ng/L or Delta hs-TnT ≥ 5 ng/L for 10/2017 to 04/2018 (hs-TnT ESC0/1h +) (Subgroup 3). (y: years; SD: standard deviation; CAG: coronary angiography; PCI: percutaneous coronary intervention; AMI: acute myocardial infarction, CHD: coronary heart disease; n.s.: not significant).

	10/2016–04/2017 (cTnI 5 × ULN/50% delta)		10/2017–04/2018 (hs-TnT ESC0/1h +)		*p*-Value
No. of cases	831		1120		
Age (y ± SD), (Min-Max)	73 ± 14	(18–102)	74 ± 13	(19–96)	0.023
Male gender, *n* (%)	473	(56.9)	701	(62.6)	<0.05
Echocardiogram (%)	742	(89.3)	978	(87.3)	n.s.
CAG (%)	444	(53.4)	520	(46.4)	<0.001
- PCI performed (% of CAG)	247	(55.6)	282	(54.2)	0.027
Mortality (%)	86	(10.3)	119	(10.6)	n.s.
AMI (%)	435	(52.3)	345	(30.8)	<0.001
CHD (%)	498	(59.9)	669	(59.7)	n.s.

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
