# Peer review of "High-Sensitivity Troponin T Testing: Consequences on Daily Clinical Practice and Effects on Diagnosis of Myocardial Infarction"

_jcm, 2020, doi:10.3390/jcm9030775_

Round 1

Reviewer 1 Report

The manuscript was carefully reviewed by reviewer#1. 

The authors addressed the concerns raised before very well. This well done large observational study what happens when hs-TnI are used in the unselected nor-ER population. There, however, is a major remaining concern. Mechanisms of decreased the ICD-10 coded of AMI is still unclear after reading rebuttal letter from the authors. This should be main message from the current manuscript. Could you explain this with use of data included into the study?

Or, if it's impossible, please provide more reasonable discussion about this topic.

Author Response

The manuscript was carefully reviewed by reviewer#1. 

The authors addressed the concerns raised before very well. This well done large observational study what happens when hs-TnI are used in the unselected nor-ER population. There, however, is a major remaining concern. Mechanisms of decreased the ICD-10 coded of AMI is still unclear after reading rebuttal letter from the authors. This should be main message from the current manuscript. Could you explain this with use of data included into the study?

Thank you for your helpful remarks. As outlined in the original manuscript, we cannot fully explain the cause for the decrease in AMI diagnosis - due to the retrospective observational design of the study.   Therefore, we strongly advocate further prospective analysis of this important issue. Our hypothesis of disregarding Type 2 MI and trivialization of pathological test results due to the 2.3-fold increase of Troponin values above the 99th percentile might help to clarify this point and is supported by the observation that even if the analysis is restricted to cases with elevated troponin and subsequent PCI, the decrease is still significant. The manuscript has been reformulated to highlight this more clearly.

Reviewer 2 Report

This bservational manuscript is better after the first revision. Still it require a lot of work to be published. The first question is the aim of this study.

This study includes almost all even already hospitalised patients (not cardiac surgery). Many of these troponin tests are taken for prognostic not diagnostic purposes. Still the main result is the number of AMI diagnoses. It important to distinguish if the troponin testing is used for diagnosis of AMI and is it done the emergency department or inhospital. This can not be found from this manuscript.

Patient is included with only one troponin test, but still the delta values for diagnosis are used. I can not find how patients with only one test are diagnosed and how many patietns had only one test. 

If I understand correctly in Table 1 is all patients (AMI 920 vs 554) and in Table 2 is hospitalised patients (AMI 739 vd 554). Why 215 (181+34) AMI patients were not hospitalised? What kind of AMI patient is not hospitalised?

The pevious problem is probably due to limitation of using only ICD coding for the diagnosis. Should all AMI diagnosis be manually checked? There might be an explanation for misdiagnosing of AMI in 2017 which is later corrected.

My opinion is that there is too many subgroubs (and tables). Because this is an observational study of Troponin use in one hospital, I suggest that only Table 1 all patients and table 3 is enough. I should include the number of AMI diagnosis in table 3 and I hope that patients that visited only in the emergency department are included in this table.

The main founding in this study is large and wide use of tropinin testing in different patient groups. First conclusion is minimal changes in the use of cardiac resources (Echo, PCI and CAG). The second is unexplained fall in the number of ICD based  AMI diagnoses.

Author Response

This observational manuscript is better after the first revision. Still it require a lot of work to be published. The first question is the aim of this study.

This study includes almost all even already hospitalised patients (not cardiac surgery). Many of these troponin tests are taken for prognostic not diagnostic purposes. Still the main result is the number of AMI diagnoses. It important to distinguish if the troponin testing is used for diagnosis of AMI and is it done the emergency department or inhospital. This can not be found from this manuscript.

Patient is included with only one troponin test, but still the delta values for diagnosis are used. I can not find how patients with only one test are diagnosed and how many patietns had only one test. 

Thank you for your insightful comments.

The present observational study represents a real world setting where only 28% of patients with cTnI testing and 27% tested with hs-TnT initially presented to the emergency department with cardiovascular complaints.

In the overall cohort, 77% of patients tested with cTnI had values below the limit of detection (LoD; cTnI 0.02 ng/ml), consecutive testing (prior to a possible coronary angiography) was performed in 35% of cases. In 20% of cases, Troponin testing showed values above the 99th percentile (cTnI 0.05ng/ml).

Regarding cases tested with hs-TnT, 16% of patients tested with hs-TnT had values below the LoD (hs-TnT 3ng/l), consecutive testing (prior to a possible coronary angiography) was performed in 40% of cases. In 47% of cases, Troponin testing showed values above the 99th percentile (hs-TnT 14 ng/l).

If I understand correctly in Table 1 is all patients (AMI 920 vs 554) and in Table 2 is hospitalised patients (AMI 739 vd 554). Why 215 (181+34) AMI patients were not hospitalised? What kind of AMI patient is not hospitalised?

We have indeed to clarify that "hospitalized" is defined by an inhospital stay > 24h, whereas 215 patients had been in the hospital for less than 24 hours. Early discharge despite positive biomarkers is not uncommon in Germany. In fact, a prospective multicenter study dealing with patients with moderate troponin levels and discharge within a 24 hours period or less without invasive imaging procedures is being conducted in Germany right now (Gray Zone Study: https://clinicaltrials.gov/ct2/show/NCT03820466). As you suggested in a comment below, we have now removed Table 2 from the manuscript.

The pevious problem is probably due to limitation of using only ICD coding for the diagnosis. Should all AMI diagnosis be manually checked? There might be an explanation for misdiagnosing of AMI in 2017 which is later corrected.

We would like to elucidate this further: In Germany a representative body of the insurance companies ("Medizinischer Dienst der Krankenkassen") independently checks patient files with healthcare cost-relevant diagnoses including myocardial infarction. As a result, overdiagnosis of myocardial infarction in either 2017 or 2018 is very unlikely due to independent surveillance.Underdiagnosis on the other hand is not questioned and leaves the problems mentioned in the discussion.

My opinion is that there is too many subgroubs (and tables). Because this is an observational study of Troponin use in one hospital, I suggest that only Table 1 all patients and table 3 is enough. I should include the number of AMI diagnosis in table 3 and I hope that patients that visited only in the emergency department are included in this table.

We accept and agree to your comment. As advised, we removed table 2 and only present the overall cohort as well as the other subgroups.

The main founding in this study is large and wide use of hs- troponin testing in different patient groups. First conclusion is minimal changes in the use of cardiac resources (Echo, PCI and CAG). The second is unexplained fall in the number of ICD based AMI diagnoses.

The main findings of this study analyzing the clinical consequences of large use of hs-testing in different patient cohorts are:

1) there are only minimal changes in the use of cardiac resources (Echo, PCI and CAG) after introduction of a more sensitive troponin test,

2) there is a fall in the number of ICD based AMI diagnoses which needs further analysis and clarification

Round 2

Reviewer 1 Report

The authors addressed the issues raised by reviewer #1. 

Reviewer 2 Report

Manuscript is better now and can be accepted.

This manuscript is a resubmission of an earlier submission. The following is a list of the peer review reports and author responses from that submission.

Round 1

Reviewer 1 Report

The manuscript entitled "High-sensitive troponin T testing: Use and consequence in daily clinical practice - a single center experience-" was carefully reviewed. The authors investigated changes in clinical practice before and after the introduction of hs-TnI testing. The concept of the present study is very unique and attractive for readers, but there are too many major concerns at this moment. 

1. Abstract:

1) The message is unclear in this context. Please, re-write logically. Study results  should support your conclusion.

2) The terms "Interestingly" or "surprisingly" are not appropriate to be used in abstract section. 

2. Introduction:

This section is well written with appropriate references. Please, clarify your study endpoints (e.g. Primary endpoints......, secondary endpoints....).

3. Experimental section:

It seems acceptable. However, the author should clarify for what purpose you conducted in this section. It would help readers understanding.

4. Results:

Major concern: the data described in result section never support your conclusion. Moreover, mechanisms of decrease in the ICD-10 coded diagnosis of AMI is unclear. 

1) As same as abstract section, the term "Interestingly" should not be used in the results section.

2) Table 1-3 should be combined and reported as one Table. It would help readers understanding. Moreover, continuous variables may be reported as mean ± SD. 

3)Do not cite reference#24 in the result section. 

Please, reconsider your study design and way of analysis in order to verify your study endpoint, logically.

Reviewer 2 Report

This well done large observational study what happens when high sensitivity troponins are used in the unselected hospitalised population.

There is not new information. It is already known that the amount of positive test results is higher. Only new finding is the obesrvation of lower level myocardial infarction diagnosis which is not explained enough. 

Still the result of only minor changes in clinical flow and use of resources is important information.

The main question what happens when high sensitivity Troponin is used instead of older tests. There is few previous reports that are very similar. This is not original topic, instead very published. This material is large single center observational study that gives not new information. Paper is well written and easy to read. Conclusions are in line with evidence and arguments. The problem of this study is that it is observational. The question they address is what happened and they answer it. The results are quite similar to previous reports. Which is a good thing, but gives not new information. If you are interested to publish this good observation, you can publish it easily. Just ask to explain why the number of AMI diagnosis decreased (clinical decision making?). Otherwise there is not much information or originality in this study.